# Impact of Metformin Use on Survival in Patients with Gastric Cancer and Diabetes Mellitus Following Gastrectomy

**DOI:** 10.3390/cancers12082013

**Published:** 2020-07-23

**Authors:** Wai-Shan Chung, Po-Hsien Le, Chiang-Jung Kuo, Tsung-Hsing Chen, Chang-Fu Kuo, Meng-Jiun Chiou, Wen-Chi Chou, Ta-Sen Yeh, Jun-Te Hsu

**Affiliations:** 1Department of General Surgery, Chang Gung Memorial Hospital at Linkou, Chang Gung University College of Medicine, Taoyuan 33305, Taiwan; m1124@cgmh.org.tw (W.-S.C.); tsy471027@cgmh.org.tw (T.-S.Y.); 2Department of Gastroenterology, Chang Gung Memorial Hospital at Linkou, Chang Gung University College of Medicine, Taoyuan 33305, Taiwan; b9005031@cgmh.org.tw (P.-H.L.); m7011@cgmh.org.tw (C.-J.K.); itochenyu@gmail.com (T.-H.C.); 3Department of Rheumatology, Allergy and Immunology, Chang Gung Memorial Hospital at Linkou, Chang Gung University College of Medicine, Taoyuan 33305, Taiwan; zandis@gmail.com; 4Center for Artificial Intelligence in Medicine, Chang Gung Memorial Hospital at Linkou, Taoyuan 33305, Taiwan; mengjiun@cgmh.org.tw; 5Department of Hematology-Oncology, Chang Gung Memorial Hospital at Linkou, Chang Gung University College of Medicine, Taoyuan 33305, Taiwan; wenchi3992@yahoo.com.tw

**Keywords:** gastric cancer, metformin, diabetes mellitus

## Abstract

Studies have shown the anticancer effects of metformin in vitro. However, whether metformin can prevent cancer recurrence or prolong survival in patients with gastric cancer (GC) and diabetes mellitus (DM) post-gastrectomy remains unknown. We evaluated the beneficial effects of metformin in patients with GC and DM post-gastrectomy. We recruited 2400 patients with GC (1749 without DM, 651 with DM) who underwent surgery between 1997 and 2010. Patients with DM were stratified into metformin (group 1) and non-metformin (group 2) users. Their clinicopathological data were recorded prospectively, and demographics, recurrence-free survival (RFS), and cancer-specific survival (CSS) were compared. Tumour recurrence risk and cause of death were analysed between groups 1 and 2 among patients with DM stratified by tumour stage. We also compared RFS and overall survival among patients with and without DM. Tumour recurrence occurred in 201 patients with GC: 57 (25%) in group 1 and 144 (37%) in group 2. After adjusting for confounders, metformin significantly prolonged CSS (hazard ratio (HR) = 0.54, 95% confidence interval (CI) = 0.38–0.77) in patients with stage I–III GC and DM. In subgroup analysis, metformin users with stage III GC and DM had significantly prolonged CSS compared to non-metformin users (HR = 0.45, 95% CI = 0.30–0.68), with an insignificant difference in patients with stage I–II GC. Adjusted HRs for RFS and CSS were significantly lower in patients with stage I–III GC and DM than those in patients without DM (0.67 (95% CI = 0.54–0.92) and 0.62 (95% CI = 0.50–0.77), respectively), with an insignificant difference in patients with stage I GC. Metformin significantly reduces tumour recurrence risk and improves CSS in patients with stage III GC and DM post-gastrectomy. Further prospective studies may confirm the efficacy of metformin as an adjunctive treatment for advanced GC postoperatively.

## 1. Introduction

Despite advances in surgical technology and multidisciplinary management, gastric cancer (GC) is the second leading cause of cancer-related deaths worldwide [1]. Even after radical surgical resection, approximately 10% to 80% of patients with GC experience relapse and die of the disease [2]. The 5-year cancer-specific survival rates among patients with advanced stage GC vary from 20% to 50% [3,4]. Type 2 diabetes mellitus (DM) has been associated with death from cancers of the liver, pancreas, colorectum, lung, bladder, ovaries, and breasts [5,6,7,8]. However, there are conflicting results about the association between DM and GC prognosis [9,10,11].

Metformin, a biguanide agent, is the most widely used and preferred first-line therapy for the treatment of DM if no contraindication exists [12]. This drug has an excellent safety profile and can be safely combined with other anti-diabetic agents [13]. Kim et al. indicated that long-term metformin use decreases GC risk, based on a Korean nationwide cohort study [14]. Recently, a retrospective study [9] from Korea showed that patients with GC without DM have better recurrence-free survival and cancer-specific survival compared with those in patients with GC and DM. However, patients with DM treated with metformin have similar survival rates to those in patients without DM and GC undergoing radical surgery, suggesting that metformin has beneficial effects on survival in patients with GC and DM.

Although basic studies have shown that metformin inhibits GC cell growth in vitro and reduces tumour spread in animal models of GC [15], few studies have addressed the beneficial effects of metformin on patients with GC and DM following surgery regarding the recurrence-free and cancer-specific survival. The aim of this study was to investigate whether metformin use reduces the risk of cancer recurrence and/or mortality in patients with stage I–III GC and DM after gastrectomy.

## 2. Materials and Methods

As shown in Figure 1, a total of 2741 patients with GC undergoing gastrectomy between 1997 and 2010 at Chang Gung Memorial Hospital at Linkou were identified. Excluding 258 patients with stage IV disease and 83 patients with in-hospital mortality, we recruited 2400 patients undergoing gastrectomy. We used the International Classification of Disease, 9th Revision, Clinical Modification codes to define DM. Data about patients with DM treated with hypoglycaemic reagents were obtained from the National Health Insurance program in Taiwan, which covers nearly the entire population of the country. Among them, 1749 patients did not have DM, and 651 had type II DM. We further stratified patients with DM into metformin and non-metformin users or patients with and without DM for comparison in terms of recurrence and overall survival. Patients with DM treated with metformin ≥ 6 months after surgery or who developed recurrence were categorised as the metformin user group.

Clinicopathological data including gender, age, comorbidity, type of operation, numbers of lymph nodes retrieved, tumour size, tumour differentiation, resection margins, pathological stage according to the 7th edition of the American Joint Cancer committee [16], presence of vascular invasion, lymphatic or perineural invasion, administration of chemotherapy, use of hypoglycaemic reagents, and cumulative time of metformin exposure were obtained from the prospectively collected electronic medical records. In order to track metformin use, we collected data on the prescriptions provided and number of days supplied. The defined daily dose (DDD) was then utilised to calculate a prescribed amount of metformin [17]. The DDD is the assumed average maintenance daily dose for the drug used. The cumulative DDD, which indicates both the dosage and duration of drug exposure, was estimated as the total amount of prescribed metformin.

The surgical procedure was mainly determined by the tumour location, pre-operative stage, and surgical findings. Radical subtotal gastrectomy for a distal third tumour and total gastrectomy for a middle or upper third tumour were performed. For selected early upper third tumours, radical proximal gastrectomy was performed with esophagogastrostomy reconstruction. Patients underwent radical surgery including D1, D1^+^, or D2 lymphadenectomy based on the depth of tumour invasion, nodal involvement status, or tumour differentiation. No patients received neoadjuvant chemotherapy. Adjuvant chemotherapy with fluoropyrimidine or platinum-based regimens was planned to be administered for suitable patients with stage II or III disease respectively, usually 6–8 weeks after surgery. Patients underwent regular follow-up with biochemistry tests or imaging studies. Recurrences were categorised as locoregional, peritoneal seeding, or hematogenous (distant). Salvage chemotherapy was planned to be administered to medically fit patients with cancer recurrence. The patient survival was calculated from the date of surgery to the date of the last follow-up (31 December 2015) or death. This study was approved by the institutional review board of Chang Gung Memorial Hospital in Taiwan (approval number 104-6943B).

All data are presented as percentages or means with standard deviations (SD). Numerical data were compared using the independent three-sample test. Pearson’s chi-square test and Fisher’s exact test were used for the analysis of nominal variables. Overall survival time (time-to-event) was calculated using the Kaplan–Meier method. We also utilised different models by adjusting for sex, age, the Charlson comorbidity index, type of gastrectomy, pathological findings, stage, and administration of chemotherapy or hypoglycaemic medications to calculate the overall survival. Statistical analyses were performed using SPSS for Windows, version 20.0 (SPSS, IBM Corp., Armonk, NY, USA). A *p*-value of <0.05 was considered significant.

## 3. Results

The clinicopathological characteristics of patients with stage I–III GC are presented in Table 1. Compared with those of the non-metformin users among patients with GC and DM, metformin users presented with a younger age (*p* = 0.0006), lower frequency of Charlson comorbidity index scores < 6 (*p* = 0.0035), higher glycohaemoglobin values (*p* < 0.0001) and glucose levels (*p* < 0.0001), greater percentage of undergoing partial gastrectomy (*p* < 0.0001), smaller tumour size (*p* = 0.0001), more differentiation (*p* = 0.0012), lower rate of positive resection margins (*p* = 0.0117), greater proportion of early-stage disease (*p* < 0.0001), less proportions of lymphatic invasion (*p* < 0.0026) or perineural invasion (*p* < 0.0009), and lower percentage of recurrence (*p* = 0.0002). There were no significant differences in gender, number of lymph nodes retrieved, presence of vascular invasion, or administration of adjuvant chemotherapy, including regimen or duration, between the 2 groups. The metformin group had higher percentages of using other hypoglycaemic reagents, except meglitinide. There were no severe adverse effects in the metformin users.

Table 2 shows the numbers of tumour recurrences and hazard ratios (HR) of deaths among patients with GC and DM stratified by stage. A total of 201 patients experienced GC recurrence, including 144 non-metformin users (37%) and 57 metformin users (25%). The metformin users accounted for significantly lower numbers of recurrence (adjusted HR = 0.61; 95% confidence interval (CI), 0.43–0.88), all-cause death (adjusted HR = 0.61; 95% CI, 0.47–0.79), and cancer-specific death (adjusted HR = 0.54; 95% CI, 0.38–0.77) as compared to those of the non-metformin group after adjusting for variable confounding factors. In subgroup analysis, only patients with stage III GC and DM treated with metformin had lower rates of recurrence (adjusted HR = 0.55; 95% CI, 0.36–083) and cancer-specific death (adjusted HR = 0.45; 95% CI, 0.30–068). Survival analysis using Kaplan–Meier plots revealed that patients with stage I–III GC and DM benefited from metformin use in terms of recurrence-free survival and cancer-specific survival with log rank *p* = 0.0002 and 0.00023, respectively (Figure 2). The beneficial effects of metformin on survival were not evident in patients with stage I or II GC and DM. The stage III metformin group had longer disease-free survival (*p* = 0.0034) and cancer-specific survival (*p* = 0.0038) as compared with the non-metformin group (Figure 3).

Patients with stage I–III GC and DM had a lower risk of recurrence (adjusted HR = 0.67; 95% CI, 0.54–082) and cancer-specific death rate compared to that in patients without DM (adjusted HR = 0.62; 95% CI, 0.50–077), as shown in Table 3. Subgroup analysis revealed that there was no difference in cancer recurrence and cancer-specific death between patients with stage I GC and DM and patients without DM. Patients with stage II–III GC and DM had lower rates of recurrence (adjusted HR = 0.51 (95% CI, 0.38–0.86) and 0.68 (95% CI, 0.53–0.87), respectively) and disease-specific death (adjusted HR = 0.36 (95% CI, 0.19–0.67) and 0.64 (95% CI, 0.50–0.82), respectively) in comparison with those in patients without DM (Table 3).

Among patients with DM, those with stage III GC treated with a metformin dose ≤ 1000 mg or > 1000 mg daily showed a reduced risk of cancer recurrence (adjusted HR = 0.46 (95% CI, 0.24–0.88) and 0.61 (95% CI, 0.39–0.94), respectively) and disease-specific death rate (adjusted HR = 0.34 (95% CI, 0.19–0.60) and 0.59 (95% CI, 0.41–0.87), respectively) compared to that in patients with stage III GC in the non-metformin group (Table 4). Figure 4 demonstrates no statistically significant difference in the outcomes of patients with stage III GC regarding the metformin dosage.

## 4. Discussion

To our best knowledge, our study is the largest-scale cohort trial conducted to evaluate the impact of metformin use on outcomes in patients with GC and DM following radical-intent gastrectomy, and shows that metformin use reduces the risk of cancer recurrence and cancer-specific death rate in patients with GC and DM. Our results are supported by the results of a retrospective study using databases of Taiwan’s National Health Insurance, indicating that metformin use significantly decreases the risk of GC [18]. In subgroup analysis, we found that metformin use in patients with stage III GC and DM is independently associated with increases in recurrence-free and cancer-specific survival after adjusting for various confounding factors to eliminate survival biases. However, the beneficial effects of metformin on survival disappear in patients with stage I or II GC. We speculate that radical surgery itself has a greater influence on prognosis than that conferred by adjuvant chemotherapy and/or hypoglycaemic agents such as metformin or insulin for patients with stage I or II GC and DM.

To further confirm the concept of salutary effects of metformin on prognosis in patients with GC, we compared the long-term outcomes between patients with GC with and without DM after surgery. Our results revealed that patients with stage II and III GC and DM have longer disease-free survival and cancer-specific survival than those in patients without DM, suggesting that hypoglycaemic agents might play a role in improving patients’ outcomes. It is reasonable to propose that metformin might reduce the risk of GC recurrence and prolong cancer-specific survival because metformin is used as a first-line therapy for patients with DM, according to the current treatment guidelines of DM [19]. In addition, studies have also shown that, compared with the use of insulin or insulin secretagogues, metformin use lowers the risk of cancer and cancer-related deaths in patients with DM [20,21].

Several studies have indicated that DM is associated with an increased risk of GC [22,23], and DM has been reported to be an independent factor of a higher risk of surgical complications [24,25]. Furthermore, Tsai et al. revealed that the risk of 90-day mortality after gastrectomy for GC is higher in patients with DM compared with that in patients without DM [26]. Severe surgical complications might delay the timing of patients with GC receiving adjuvant chemotherapy, which would subsequently influence patient outcomes. In contrast to the aforementioned findings, favourable prognosis in patients with DM or metformin users after surgery compared with that in patients without DM or non-metformin users was observed in the present study. Moreover, studies have shown that insulin can promote cancer growth [27,28], and our data revealed that even though there were more patients in the metformin group concomitantly treated with insulin compared with those in the non-metformin group (60% vs. 31%), improved survival among the metformin users was still identified after adjusting for other confounding determinants, implying that metformin might prevent the recurrence or progression of GC. Similar findings were also reported by Lee et al. in a Korean population [9]. An additional basic study also indicated that metformin enhances the cisplatin-induced reduction in GC cell growth in mice [29]. In the current study, patients with stage III disease usually received platinum-based adjuvant chemotherapy. Collectively, the primary anticancer effect of metformin is probably attributed to modulating various molecular pathways in GC rather than improving survival by controlling DM.

Metformin improves blood glucose levels by increasing insulin sensitivity and reducing gluconeogenesis by the liver, which helps in diminishing insulin production and therefore decreases the risk of cancer development [30]. Basic studies have indicated that metformin inhibits cancer cell growth through inducing cell cycle arrest and promoting apoptosis [31,32,33,34]. The anticancer effects of metformin occur through reducing circulating insulin levels and inhibiting insulin and insulin-like growth factor 1 signalling pathways, which are involved in the regulation of glucose uptake and carcinogenesis [35,36]. Studies suggest that metformin inhibits mammalian target of rapamycin (mTOR)-dependent and independent AMP-activated protein kinase (AMPK) activation in breast cancer or GC cells [29,33,34]. Mohammed et al. reported that metformin reduces pancreatic carcinoma metastasis in transgenic mice with significant inhibition of mTOR and an increase in phosphorylated AMPK and TSC2 [37]. Kato et al. demonstrated that metformin inhibits the growth of GC cells by reducing cyclin D1, cyclin-dependent kinase (Cdk) 4, and Cdk 6 in vitro and in vivo [15]. Nonetheless, the precise molecular mechanism of the anticancer effects of metformin remains unclear and under investigation.

Kim et al. also demonstrated that long-term metformin use lowers the risk of GC in patients with DM without insulin treatment based on a Korean nationwide cohort study [14]. Clinically, only one study conducted by Lee et al., in which they recruited small numbers of patients with stage III disease, revealed that the survival benefits of cumulative metformin use in patients with GC and DM after surgery are observed in patients with stage III but not stage I or II disease [9]. After adjusting for confounding factors, for each additional cumulative 6 months of metformin use following gastrectomy, there is a 13.6% reduction in the risk of cancer recurrence (HR = 0.864; 95% CI, 0.797–0.937) and a 13.5% decrease in the cancer-specific mortality rate (HR = 0.865; 95% CI, 0.782–0.958) [9]. In line with their findings, we enrolled a large-scale cohort, stratified patients with GC and DM according to cancer stage for analysis, and found that metformin use significantly prolongs recurrence-free and cancer-specific survival rates in patients with stage III GC and DM following surgery, with 45.0% and 55.0% decreases in the risk of tumour recurrence (adjusted HR = 0.55; 95% CI, 0.36–083) and cancer-specific mortality rate (adjusted HR = 0.45; 95% CI, 0.30–068), respectively (Table 3).

## 5. Limitations

Several concerns and limitations exist in the present study. First, this was an observational study, and we were unable to explain the underlying carcinogenic mechanism involving DM, GC, and metformin. Second, the database did not include insulin levels, and the patients’ performance statuses were not explicitly considered in the data analysis. Third, drug compliance is likely to cause a misrepresentation of the medication’s effect. Nonetheless, our results are likely to demonstrate significantly beneficial effects of metformin use for patients with advanced GC and DM. Furthermore, to evaluate whether patients with GC without DM will benefit from metformin use, a prospective and randomised trial is required.

## 6. Conclusions

Metformin use significantly reduces the risk of tumour recurrence and confers cancer-specific survival benefits in patients with stage III GC and DM after surgery. Further prospective studies are needed to validate the efficacy of metformin as an adjunctive treatment in patients with advanced GC following surgery.

## Figures and Tables

**Figure 1 cancers-12-02013-f001:**
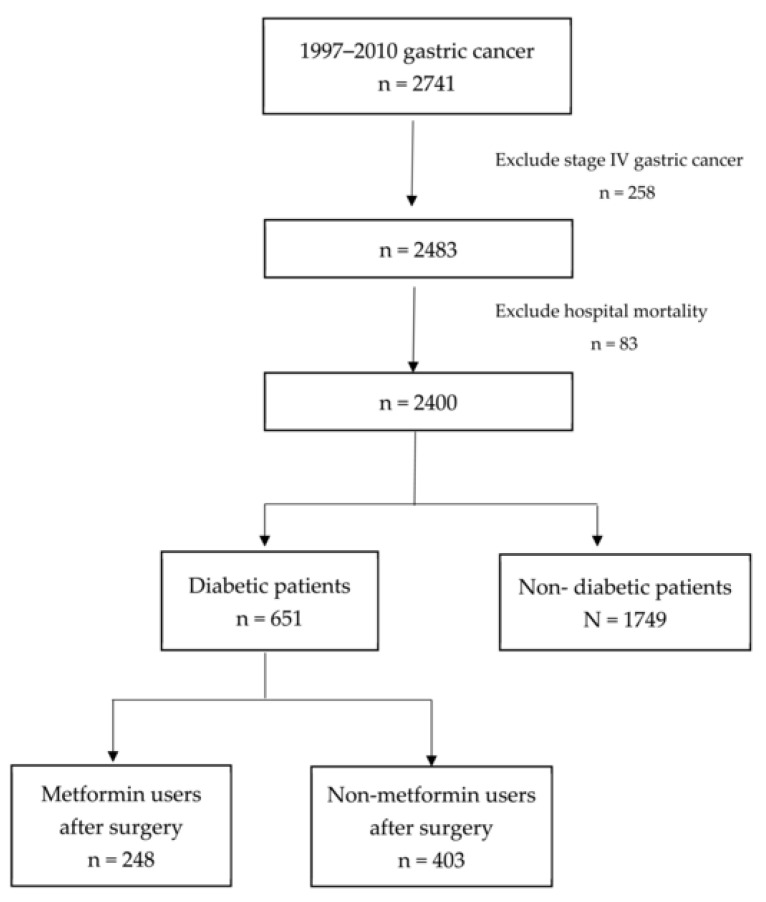
Flowchart of patients with stage I–III gastric cancer (GC) undergoing gastrectomy recruited in the study.

**Figure 2 cancers-12-02013-f002:**
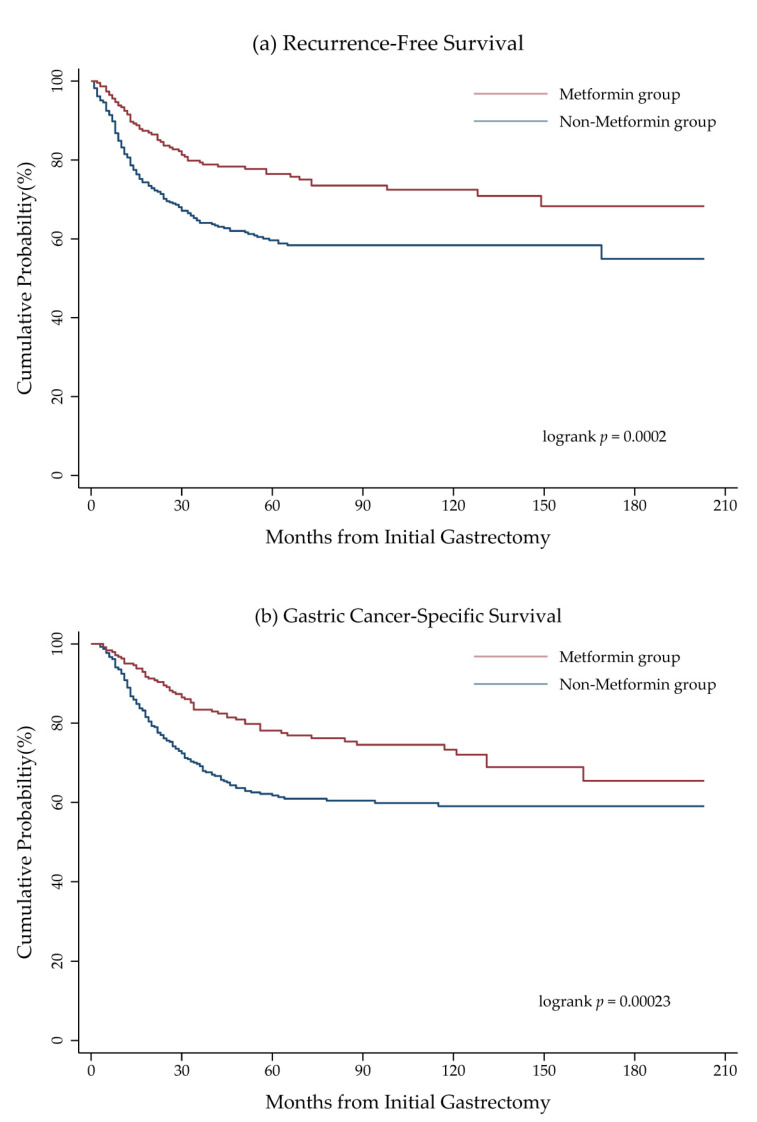
(**a**) Recurrence-free survival and (**b**) cancer-specific survival in patients with stage I–III GC and DM in terms of metformin use.

**Figure 3 cancers-12-02013-f003:**
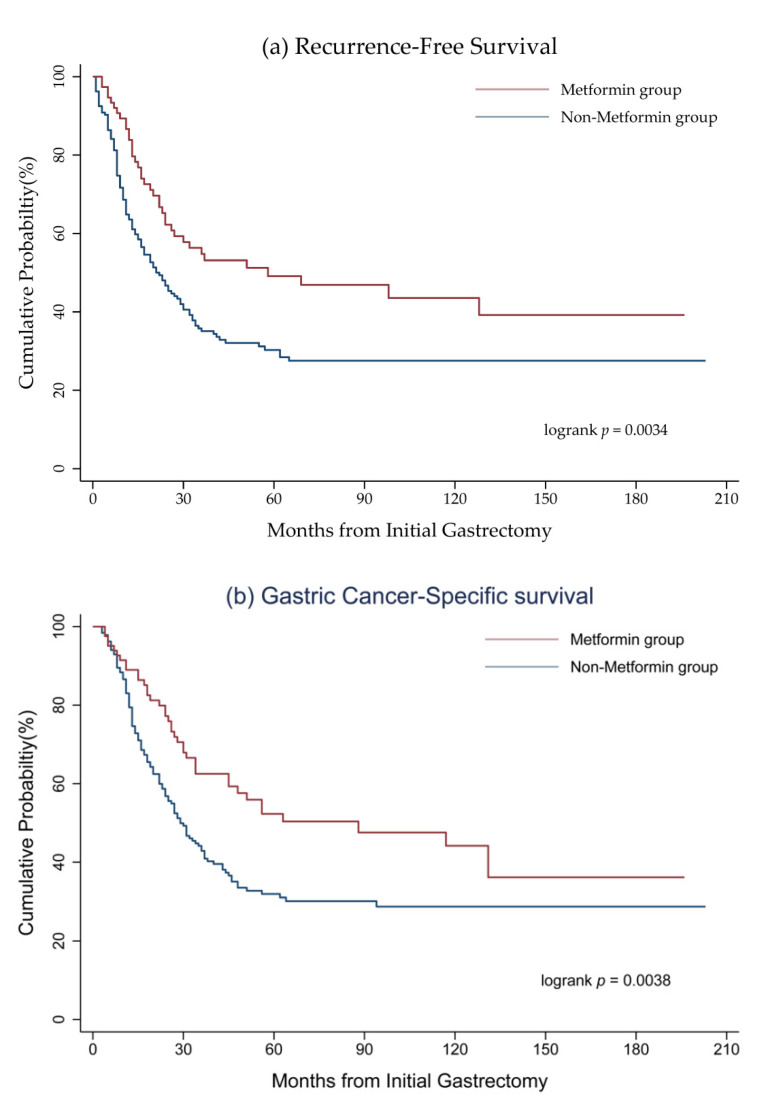
(**a**) Recurrence-free survival and (**b**) cancer-specific survival in patients with stage III GC and DM in terms of metformin use.

**Figure 4 cancers-12-02013-f004:**
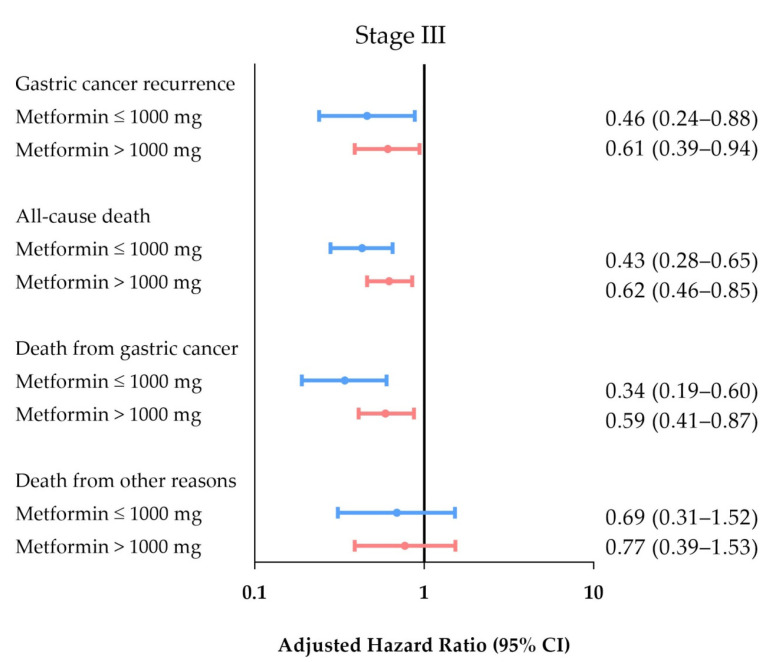
Impact of metformin dosage on recurrence-free survival and survival in patients with stage III GC and DM.

**Table 1 cancers-12-02013-t001:** Clinicopathological characteristics of patients with stage I-III GC.

Parameters	Total Patients *n* = 2400		Patients with Diabetes (*n* = 651)				*p*-Value
			Metformin Users *n* = 248		Non-Metformin Users *n* = 403		
Gender							0.8260
Male	1515	(63.13)	148	(59.68)	244	(60.55)	
Female	885	(36.88)	100	(40.32)	159	(39.45)	
Age (Years)							0.0006
Mean ± Standard Deviation	63.37 ± 13.17		65.49 ± 10.48		68.50 ± 10.94		
Charlson Comorbidity Index Score							
Mean ± Standard Deviation	5.42 ± 3.20		5.69 ± 3.43		6.03 ± 3.45		0.2090
≤6	1333	(55.54)	160	(64.52)	213	(52.85)	0.0035
>6	1067	(44.46)	88	(35.48)	190	(47.15)	
Glycohaemoglobin (HbA1c, %)							<0.0001
Mean ± Standard Deviation			7.34 ± 1.39		6.33 ± 1.01		
Glucose, AC (mg/dL)							<0.0001
Mean ± Standard Deviation			149.00 ± 49.27		116.33 ± 40.9		
Type of Gastrectomy							<0.0001
Total Gastrectomy	646	(26.92)	37	(14.92)	122	(30.27)	
Partial Gastrectomy	1754	(73.08)	211	(85.08)	281	(69.73)	
Number of Lymph Node Retrieved							0.8016
Mean ± Standard Deviation	29.81 ± 16.07		29.17 ± 17.36		28.84 ± 15.94		
Tumour Size (cm)							0.0001
Mean ± Standard Deviation	4.30 ± 2.96		3.49 ± 2.38		4.32 ± 3.00		
Differentiation							0.0012
Yes	1395	(58.13)	111	(44.76)	233	(57.82)	
No	1005	(41.88)	137	(55.24)	170	(42.18)	
Resection Margins							0.0117
Negative	2236	(93.17)	242	(97.58)	375	(93.05)	
Positive	164	(6.83)	6	(2.42)	28	(6.95)	
Stage (AJCC 7th ed.)							<0.0001
I	746	(31.08)	116	(46.77)	123	(30.52)	
II	460	(19.17)	49	(19.76)	88	(21.84)	
III	1194	(49.75)	83	(33.47)	192	(47.64)	
Vascular Invasion							0.2553
No	2091	(87.13)	224	(90.32)	361	(89.58)	
Yes	269	(11.21)	19	(7.66)	39	(9.68)	
Unknown	40	(1.67)	5	(2.02)	3	(0.74)	
Lymphatic Invasion							0.0026
No	1197	(49.88)	154	(62.10)	200	(49.63)	
Yes	1168	(48.67)	89	(35.89)	199	(49.38)	
Unknown	35	(1.46)	5	(2.02)	4	(0.99)	
Perineural Invasion							0.0009
No	1385	(57.71)	175	(70.56)	232	(57.57)	
Yes	975	(40.63)	67	(27.02)	166	(41.19)	
Unknown	40	(1.67)	6	(2.42)	5	(1.24)	
Adjuvant Chemotherapy							0.5224
No	951	(39.63)	119	(47.98)	183	(45.41)	
Yes	1449	(60.38)	129	(52.02)	220	(54.59)	
Chemotherapy Regimen							0.6432
Fluoropyrimidine-Based	1338	(95.79)	126	(97.67)	211	(95.91)	
Platinum-Based	50	(3.45)	2	(1.55)	7	(3.18)	
Others	11	(0.76)	1	(0.78)	2	(0.91)	
Chemotherapy Duration (Months)							
Mean ± Standard Deviation	7.98 ± 6.74		8.10 ± 6.87		7.92 ± 6.68		0.8340
Other Hypoglycaemic Medication							
Sulfonylurea	346	(14.42)	174	(70.16)	150	(37.22)	<0.0001
Meglitinide	37	(1.54)	19	(7.66)	18	(4.47)	0.0873
α-Glucosidase Inhibitor	42	(1.75)	25	(10.08)	17	(4.22)	0.0031
Thiazolidinedione	52	(2.17)	33	(13.31)	19	(4.71)	<0.0001
Insulin	320	(13.33)	149	(60.08)	124	(30.77)	<0.0001
Recurrence Pattern							0.0002
No	1412	(58.83)	191	(77.02)	259	(64.27)	
Locoregional	235	(23.79)	22	(8.87)	27	(6.70)	
Peritoneal	355	(35.93)	14	(5.65)	53	(13.15)	
Hemaetogenous	398	(40.28)	21	(8.47)	64	(15.88)	

Values are numbers, with percentages in parentheses, unless otherwise stated. AC, Ante Cibum (before meals); AJCC, American Joint Cancer committee.

**Table 2 cancers-12-02013-t002:** Recurrence and death in patients with gastric cancer (GC) and diabetes mellitus (DM) treated with and without metformin, stratified by stage.

Prognosis	Stage I–III		Stage I		Stage II		Stage III	
	Non-Metformin Users (*n* = 403)	Metformin Users (*n* = 248)	Non-Metformin Users (*n* = 123)	Metformin Users (*n* = 116)	Non-Metformin Users (*n* = 88)	Metformin Users (*n* = 49)	Non-Metformin Users (*n* = 192)	Metformin Users (*n* = 83)
Gastric Cancer Recurrence								
Number of Patients	144 (37.02) ^a^	57 (24.89) ^a^	8 (6.78) ^b^	9 (8.49) ^b^	21 (25.00) ^c^	10 (21.28) ^c^	115 (61.50) ^d^	38 (50.00) ^d^
Crude HR (95% CI)	1	0.60 (0.45–0.82) *	1	1.27 (0.49–3.27)	1	0.84 (0.40–1.76)	1	0.68 (0.48–0.96) *
Adjusted HR (95% CI) ^e^	1	0.61 (0.43–0.88) *	1	1.45 (0.32–6.58)	1	0.44 (0.16–1.21)	1	0.55 (0.36–0.83) *
All-Cause Death								
Number of Patients	251 (62.28)	119 (47.98)	48 (39.02)	35 (30.17)	45 (51.14)	24 (48.98)	158 (82.29)	60 (72.29)
Crude HR (95% CI)	1	0.63 (0.51–0.79) *	1	0.74 (0.48–1.14)	1	0.90 (0.55–1.47)	1	0.64 (0.47–0.86) *
Adjusted HR (95% CI) ^e^	1	0.61 (0.47–0.79) *	1	0.69 (0.37–1.28)	1	0.95 (0.45–2.00)	1	0.52 (0.38–0.73) *
Death from Gastric Cancer								
Number of Patients	139 (34.49)	59 (23.79)	9 (7.32)	8 (6.90)	17 (19.32)	11 (22.45)	113 (58.85)	40 (48.19)
Crude HR (95% CI)	1	0.57 (0.42–0.77) *	1	0.90 (0.35–2.33)	1	1.09 (0.51–2.34)	1	0.59 (0.41–0.85) *
Adjusted HR (95% CI) ^e^	1	0.54 (0.38–0.77) *	1	1.70 (0.45–6.47)	1	0.47 (0.16–1.40)	1	0.45 (0.30–0.68) *
Death from Other Reasons								
Number of Patients	112 (27.79)	60 (24.19)	39 (31.71)	27 (23.28)	28 (31.82)	13 (26.53)	45 (23.44)	20 (24.10)
Crude HR (95% CI)	1	0.71 (0.52–0.97) *	1	0.74 (0.45–1.21)	1	0.73 (0.38–1.42)	1	0.76 (0.44–1.29)
Adjusted HR (95% CI) ^e^	1	0.75 (0.51–1.11)	1	0.53 (0.27–1.03)	1	2.35 (0.99–5.58)	1	0.71 (0.37–1.36)

Values are numbers, with percentages in parentheses, unless otherwise stated. New-onset diabetic patients after tumour recurrence treated with metformin were classified to the metformin-user group and were included into analysis of cause of death. CI, confidence interval; HR, hazard ratio; *, *p* < 0.05. ^a^ Between surgery and recurrence: 389 non-metformin users and 229 metformin users. ^b^ Between surgery and recurrence: 118 non-metformin users and 106 metformin users. ^c^ Between surgery and recurrence: 84 non-metformin users and 47 metformin users. ^d^ Between surgery and recurrence: 187 non-metformin users and 76 metformin users. ^e^ Adjusted for sex, age, Charlson comorbidity index score, type of gastrectomy, number of lymph nodes retrieved, tumour size, differentiation, resection margins, stage, vascular invasion, lymphatic invasion, perineural invasion, chemotherapy, and hypoglycaemic medications.

**Table 3 cancers-12-02013-t003:** Recurrence and death in GC patients with and without DM, stratified by stage.

Prognosis	Stage I–III		Stage I		Stage II		Stage III	
	without DM (*n* = 1749)	with DM (*n* = 651)	without DM (*n* = 507)	with DM (*n* = 239)	without DM (*n* = 323)	with DM (*n* = 137)	without DM (*n* = 919)	with DM (*n* = 275)
Gastric Cancer Recurrence								
Number of Patients	787 (44.16) ^a^	201 (32.52) ^a^	37 (7.09%) ^b^	17 (7.59) ^b^	99 (30.09) ^c^	31 (23.66) ^c^	651 (69.92) ^d^	153 (58.17) ^d^
Crude HR (95% CI)	1	0.67 (0.57–0.78) *	1	1.06 (0.60–1.88)	1	0.77 (0.52–1.16)	1	0.71 (0.60–0.84) *
Adjusted HR (95% CI) ^e^	1	0.67 (0.54–0.82) *	1	0.94 (0.48–1.86)	1	0.51 (0.30–0.86) *	1	0.68 (0.53–0.87) *
All-Cause Death								
Number of Patients	1047 (59.86)	370 (56.84)	152 (29.98)	83 (34.73)	151 (46.75)	69 (50.36)	744 (80.96)	218 (79.27)
Crude HR (95% CI)	1	0.90 (0.80–1.01)	1	1.22 (0.93–1.59)	1	1.10 (0.83–1.47)	1	0.93 (0.80–1.08)
Adjusted HR (95% CI) ^e^	1	0.70 (0.59–0.82) *	1	0.78 (0.55–1.12)	1	0.51 (0.35–0.74) *	1	0.73 (0.59–0.90) *
Death from Gastric Cancer								
Number of Patients	760 (43.45)	198 (30.41)	31 (6.11)	17 (7.11)	91 (28.17)	28 (20.44)	638 (69.42)	153 (55.64)
Crude HR (95% CI)	1	0.66 (0.57–0.78) *	1	1.18 (0.65–1.13)	1	0.76 (0.50–1.16)	1	0.76 (0.64–0.91) *
Adjusted HR (95% CI) ^e^	1	0.62 (0.50–0.77) *	1	0.97 (0.47–2.02)	1	0.36 (0.19–0.67) *	1	0.64 (0.50–0.82) *
Death from Other Reasons								
Number of Patients	287 (16.41)	172 (26.42)	121 (23.87)	66 (27.62)	60 (18.58)	41 (29.93)	106 (11.53)	65 (23.64)
Crude HR (95% CI)	1	1.50 (1.24–1.81) *	1	1.23 (0.91–1.65)	1	1.61 (1.08–2.39) *	1	1.89 (1.39–2.58) *
Adjusted HR (95% CI) ^e^	1	0.86 (0.67–1.10)	1	0.73 (0.48–1.09)	1	0.64 (0.38–1.09)	1	1.26 (0.83–1.92)

Values are numbers, with percentages in parentheses, unless otherwise stated. New-onset diabetic patients after tumour recurrence were classified to the diabetic group and were included into analysis of cause of death. CI, confidence interval; HR, hazard ratio; *, *p* < 0.05. ^a^ Between surgery and recurrence: 1782 non-diabetics and 618 diabetics. ^b^ Between surgery and recurrence: 534 non-diabetics and 232 diabetics. ^c^ Between surgery and recurrence: 344 non-diabetics and 138 diabetics. ^d^ Between surgery and recurrence: 960 non-diabetics and 275 diabetics. ^e^ Adjusted for sex, age, Charlson comorbidity index score, type of gastrectomy, number of lymph nodes retrieved, tumour size, differentiation, resection margins, stage, vascular invasion, lymphatic invasion, perineural invasion, chemotherapy, and hypoglycaemic medications.

**Table 4 cancers-12-02013-t004:** Impact of metformin dosage on recurrence and death in patients with stage III GC and DM.

Prognosis	Non-Metformin Users (*n* = 192)	Metformin ≤ 1000 mg/day (*n* = 32)	Metformin > 1000 mg/day (*n* = 51)
Gastric Cancer Recurrence			
Number of Patients	115 (61.50) ^a^	12 (41.38) ^a^	26 (55.32) ^a^
Crude HR (95% CI)	1	0.52 (0.30–0.90) *	0.79 (0.53–1.18)
Adjusted HR (95% CI) ^b^	1	0.46 (0.24–0.88) *	0.61 (0.39–0.94) *
All-Cause Death			
Number of Patients	158 (82.29)	20 (62.50)	40 (78.43)
Crude HR (95% CI)	1	0.51 (0.32–0.81) *	0.73 (0.52–1.03)
Adjusted HR (95% CI) ^b^	1	0.43 (0.28–0.65) *	0.62 (0.46–0.85) *
Death from Gastric Cancer			
Number of Patients	113 (58.85)	11 (34.38)	29 (56.86)
Crude HR (95% CI)	1	0.39 (0.21–0.73) *	0.74 (0.49–1.11)
Adjusted HR (95% CI) ^b^	1	0.34 (0.19–0.60) *	0.59 (0.41–0.87) *
Death from Other Reasons			
Number of Patients	45 (23.44)	9 (28.13)	11 (21.57)
Crude HR (95% CI)	1	0.82 (0.40–1.68)	0.72 (0.37–1.39)
Adjusted HR (95% CI) ^b^	1	0.69 (0.31–1.52)	0.77 (0.39–1.53)

Values are numbers, with percentages in parentheses, unless otherwise stated. New-onset diabetic patients after tumour recurrence treated with metformin were classified to the metformin-user groups and were included into analysis of cause of death. CI, confidence interval; HR, hazard ratio; * *p* < 0.05. ^a^ Between surgery and recurrence: 187 non-metformin users, 29 metformin users with dose ≤ 1000 mg/day and 47 metformin users with dose > 1000 mg/day. ^b^ Adjusted for sex, age, Charlson comorbidity index score, type of gastrectomy, number of lymph nodes retrieved, tumour size, differentiation, resection margins, stage, vascular invasion, lymphatic invasion, perineural invasion, chemotherapy, and hypoglycaemic medications.

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
