# Peer review of "Impact of Metformin Use on Survival in Patients with Gastric Cancer and Diabetes Mellitus Following Gastrectomy"

_cancers, 2020, doi:10.3390/cancers12082013_

Round 1
Reviewer 1 Report
This paper presents interesting data concerning the impact of metformin in gastric cancer (GC) patients with DM. I would be very interested to see the effect of metformin in GC patients without DM. The paper shows the clear limitations of the study but the level of glucose in the blood should be presented if possible since it has been described that the anti-tumoral effect of metformin can be affected by the glycemic state therefore should be taken in during hypoglycemia.
https://www.sciencedirect.com/science/article/pii/S1535610819301527
Author Response
Reviewer #1:
This paper presents interesting data concerning the impact of metformin in gastric cancer (GC) patients with DM. I would be very interested to see the effect of metformin in GC patients without DM. The paper shows the clear limitations of the study but the level of glucose in the blood should be presented if possible since it has been described that the anti-tumor effect of metformin can be affected by the glycemic state therefore should be taken in during hypoglycemia.
Response: We thank the reviewer for this important remark. We totally agree with the reviewer that that glucose levels might affect anti-tumor effect of metformin. We have added the information including glycohematoglobulin (HbA1c) and glucose levels in the table 1 and section of Results in the revised manuscript.
Reviewer 2 Report
Chung et al. evaluated the effect of metformin on recurrence-free survival and cancer-specific survival in patients with gastric cancer and diabetes mellitus who underwent surgical - and to some extent - adjuvant systemic treatment.
Comments:
- Why was the last follow up conducted 7 years ago? Since this is a prospective database, what have happened with it in the past decade? (last gastrectomy in 2010)
- There is no specific data on the reccurence - local? regional (nodal)? peritoneal?
- In my humble opinion, the Discussion is the best section of the manuscript - The role of DM and metformin in cancer has been explained extensively and smoothly.
- I appreciate the fact, that you realize the limitations of your study. I belive that manuscript would benefit if you could add data regarding the 2nd limitations - assessment of severity of DM
- In Table 1 the information regarding chemotherapy should be changed (from "chemotherapy" to "adjuvant chemotherapy" since there was no neoadjuvant treatment in your cohort)
- Table 2 and 3 are hardly understandable for an average reader - please try to re-arrange them
- Figure 2 and 3 should be of better quality - the layout seem repulsive
After conducting a major revision, the manuscript can be re-assessed for publication in Cancers.
Author Response
Reviewer #2:
- Why was the last follow up conducted 7 years ago? Since this is a prospective database, what have happened with it in the past decade? (last gastrectomy in 2010)
Responses: For the concern of enough follow-up time, we recruited the last patient in 2010. The date of the last follow-up has been corrected as December 31, 2015 for calculating survival (corrected). This study was approved by the institutional review board of Chang Gung Memorial Hospital in Taiwan on Sep 22, 2015 (approval number 104-6943B). We initiated this study in 2016. For personal reasons, we did not finish manuscript writing/editing in until this June.
- There is no specific data on the recurrence - local? regional (nodal)? peritoneal?
Responses: We have added recurrence data and sentences to the table 1 and section of Materials/Methods and Results, respectively in the revised manuscript.
- In my humble opinion, the Discussion is the best section of the manuscript - The role of DM and metformin in cancer has been explained extensively and smoothly. I appreciate the fact, that you realize the limitations of your study. I believe that manuscript would benefit if you could add data regarding the 2nd limitations - assessment of severity of DM.
Responses: We thank greatly the reviewer’s remarks that the role of DM and metformin in cancer has been explained extensively and smoothly in the discussion. We also agree very much with the reviewer’s comments that additional data in assessment of DM severity is helpful in increasing scientific level of this manuscript. We have added the data including glycohematoglobulin (HbA1c) and glucose levels to the table 1 in the revised manuscript. The sentence of 2nd limitation was modified to “Second, the database did not include insulin levels, and the patients’ performance statuses were not explicitly considered in the data analysis.”
- In Table 1 the information regarding chemotherapy should be changed (from "chemotherapy" to "adjuvant chemotherapy" since there was no neoadjuvant treatment in your cohort)
Responses: We have changed “chemotherapy “ to adjuvant chemotherapy in the revised table 1.
- Table 2 and 3 are hardly understandable for an average reader - please try to re-arrange them
Responses: We have modified tables 2 and 3 in the revised manuscript to make it easily read.
- Figure 2 and 3 should be of better quality - the layout seem repulsive
Responses: We replaced figures 2 and 3 with new ones with good quality and high resolution in the revised manuscript.
Reviewer 3 Report
Title: Impact of metformin use on survival in patients with gastric cancer and diabetes mellitus following gastrectomy
Comments to the editor
Thank you for giving me the chance to review this manuscript. This observational study demonstrated that the beneficial effects of metformin for patients with gastric cancer and diabetes mellitus. The authors concluded that metformin use reduces the risk of tumor recurrence and confers cancer-specific survival benefits in patients with stage III gastric cancer and diabetes mellitus.
This is well written study and novel information, however, this study has some limitations, and my judge is “Minor revise”
Comments to the author
This observational study demonstrated that the beneficial effects of metformin for patients with gastric cancer and diabetes mellitus.
This is well written study and novel information, however, this study has some limitations in the following areas.
Minor
- The recruited patients in this study have been underwent gastrectomy. I think it is better to specify that total 2741 patients with gastric cancer has been undergoing gastrectomy also in figure 1, because endoscopic treatment is one option for early gastric cancer.
- Adjuvant chemotherapy can be considered for patients with stage II or III gastric cancer. Regimen of chemotherapy or whether patients could be received chemotherapy for standard period may influence prognosis of gastric cancer. The authors would better to describe if there is a difference in regimen or duration of chemotherapy between metformin users and not-metformin users.
- Were there any patients considered to have been accompanying an adverse effect by metformin? Please describe it.
- I am concerned that metformin users in this study presented with lower stage of gastric cancer. Is this result also influenced by metformin use? If it is possible, please describe your opinion about that.
Author Response
Reviewer #3:
This is well written study and novel information, however, this study has some limitations in the following areas.
Responses: We gratefully thank for the reviewer’s comments that our manuscript is well written and provides novel information.
- The recruited patients in this study have been underwent gastrectomy. I think it is better to specify that total 2741 patients with gastric cancer has been undergoing gastrectomy also in figure 1, because endoscopic treatment is one option for early gastric cancer.
Responses: We thank the reviewer for this important remark. We have mentioned this point in the section of Materials and Methods. “As shown in Figure 1, a total of 2,741 patients with GC undergoing gastrectomy between 1997 and 2010 at Chang Gung Memorial Hospital at Linkou were identified.” We also revised the legend in figure 1 to “Flowchart of patients with stage I–III gastric cancer undergoing gastrectomy recruited in the study.”
- Adjuvant chemotherapy can be considered for patients with stage II or III gastric cancer. Regimen of chemotherapy or whether patients could be received chemotherapy for standard period may influence prognosis of gastric cancer. The authors would better to describe if there is a difference in regimen or duration of chemotherapy between metformin users and non-metformin users.
Responses: We thank for the reviewer’s comments. We have added the information in the table 1 and section of Results indicating that there were no significant difference in the chemotherapy regimen or duration between metformin users and non-metformin users.
- Were there any patients considered to have been accompanying an adverse effect by metformin? Please describe it.
Responses: Metformin is an old dug and has an excellent safety profile even combined with other anti-diabetic agent, and has been widely used and preferred first-line therapy for the treatment of DM. We did not find severe adverse effects in the metformin user group. We added the statement that “There were no severe adverse effects in the metformin users.” in the section of Results.
- I am concerned that metformin users in this study presented with lower stage of gastric cancer. Is this result also influenced by metformin use? If it is possible, please describe your opinion about that.
Responses: We thank for the reviewer’s remark. The unbalanced clinicopathological features including the stage between metformin and non-metformin users have been adjusted for analyzing the outcomes, shown in the footnote of table 2. (e Adjusted for sex, age, Charlson comorbidity index score, type of gastrectomy, number of lymph node retrieved, tumor size, differentiation, resection margins, stage, vascular invasion, lymphatic invasion, perineural invasion, chemotherapy and hypoglycemic medications.) This has also been mentioned in the section of Results. The metformin users accounted for significantly lower numbers of recurrence (adjusted HR=0.61; 95% CI, 0.43–0.88) and cancer-specific death (adjusted HR=0.54; 95% CI, 0.38–0.77) as compared to those of the non-metformin group after adjusting for variable confounding factors.
Round 2
Reviewer 2 Report
The authors have adressed all comments adequately. Recommend manuscript acceptance in present form.